# Postoperative swallowing recovery in oral and oropharyngeal cancer: A prospective analysis of functional changes and adjuvant therapy effects

**Loan Thi Hong Nguyen**[1], **Duc Tan Vo**[2,3], **Truc Thanh Thai**[4], **Xuan Quang Ly**[1,5]*

1 Department of Otolaryngology, Head and Neck Surgery, School of Medicine, University of Medicine and Pharmacy at Ho Chi Minh City, Ho Chi Minh City, Vietnam, 2 Department of Radiology, School of Medicine, University of Medicine and Pharmacy at Ho Chi Minh City, Ho Chi Minh City, Vietnam, 3 Department of Radiology, University Medical Center Ho Chi Minh City, Ho Chi Minh City, Vietnam, 4 Department of Medical Statistics and Informatics, University of Medicine and Pharmacy at Ho Chi Minh City, Ho Chi Minh City, Vietnam, 5 Department of Otolaryngology, Head and Neck Surgery, University Medical Center Ho Chi Minh City, Ho Chi Minh City, Vietnam

* quang.lx@umc.edu.vn

## Abstract

Dysphagia is a prevalent and debilitating sequela in patients with oral cavity and oropharyngeal cancers undergoing surgery, often complicated by adjuvant radiotherapy or chemoradiotherapy. This prospective cohort study aimed to describe the longitudinal changes in swallowing and oral intake and assess the influence of adjuvant treatment modalities. We included 89 patients with oral cavity or oropharyngeal squamous cell carcinoma. Swallowing was assessed at 1, 3, and 6 months post-surgery using Eating Assessment Tool-10 (EAT-10), Swallowing Ability and Safety Scale (SASS), Functional Oral Intake Scale (FOIS), and Fiberoptic Endoscopic Evaluation of Swallowing (FEES). Patients were stratified by adjuvant treatment: none, radiotherapy, or chemoradiotherapy. Swallowing recovery was dynamic. EAT-10 scores increased from 8.8 at 1 month to peak at 13.5 at 3 months (p < 0.001), before decreasing to 8.3 at 6 months (p < 0.001). Penetration/aspiration for thin liquids significantly increased from 32.6% at 1 month to 44.9% at 3 months (p = 0.034), then decreased to 31.5% at 6 months (p = 0.014). FOIS and SASS scores showed overall improvement from 1 to 6 months (p < 0.001 for both), despite a transient dip in FOIS and a rise in poor SASS scores at 3 months. Patients receiving chemoradiotherapy demonstrated greater perceived dysphagia, lower functional oral intake, and higher rates of thin liquid penetration/aspiration compared to other groups across all time points. In conclusion, chemoradiotherapy is associated with more severe and prolonged dysphagia. These findings underscore the critical need for targeted rehabilitation and comprehensive multidisciplinary care.

**Data availability statement:** Dataset is avalable in the appendix.

**Funding:** The author(s) received no specific funding for this work.

**Competing interests:** The authors have declared that no competing interests exist.

## Introduction

Oral cavity and oropharyngeal cancers represent a significant global health burden. Treatment strategies for these malignancies have evolved considerably, often involving complex surgical resections, frequently followed by adjuvant radiotherapy or chemoradiotherapy, aimed at achieving optimal oncological control [1–3]. While these aggressive multimodal therapies have improved survival rates, they often come at the cost of substantial functional impairments, profoundly impacting patients' quality of life [3–6].

One of the most debilitating sequelae of oral cavity and oropharyngeal cancers and their treatment is dysphagia [2,3,6]. Dysphagia in head and neck cancer (HNC) patients is multifactorial, arising from the tumor itself, surgical tissue removal, and the acute and late toxicities induced by radiation and chemotherapy, such as mucositis, xerostomia, fibrosis, and neuropathies [2,3,6–8]. The consequences of dysphagia are far-reaching, leading to malnutrition, dehydration, aspiration pneumonia, reduced social engagement, and diminished overall health-related quality of life [3,7].

Given the profound impact of dysphagia, comprehensive assessment and effective rehabilitation are paramount in the long-term management of these patients [2,3,9]. Swallowing outcomes can be evaluated through various paradigms, including patient-reported outcome measures that capture subjective experiences of difficulty and quality of life, as well as objective clinical assessments of swallowing ability and functional oral intake [10–15]. While acute dysphagia often manifests during or shortly after treatment, late-onset or persistent swallowing difficulties can emerge months to years post-treatment, emphasizing the need for longitudinal follow-up [2,3,7,8,16].

Despite growing recognition of dysphagia, the precise longitudinal trajectories of swallowing recovery and the differential impacts of various adjuvant treatment modalities on these trajectories, particularly when comparing patient-reported and objective outcomes, remain areas of ongoing investigation. Understanding these patterns is critical for accurate patient counseling, setting realistic expectations, and designing tailored rehabilitation strategies to mitigate long-term morbidity [3]. In Vietnam, to date, there has been only one study on swallowing training in patients after tongue and floor of mouth cancer surgery [17]. However, the impact of treatment often results in significant long-term functional deficits. To date, limited research has focused on the longitudinal changes in swallowing outcomes, especially when comparing subjective and objective metrics during the acute post-treatment phase. Recent large cohort studies highlight that these functional deficits, notably dysphagia, can worsen and persist for up to five years post-treatment [18], underscoring a critical need to monitor these complex trajectories starting from the early period.

Therefore, this study aims to comprehensively describe the longitudinal changes in both patient-reported and objectively assessed swallowing and oral intake outcomes in patients undergoing surgical treatment for oral cavity and oropharyngeal cancer, specifically focusing on their trajectories during the first 6 months post-surgery. Furthermore, it seeks to investigate how different adjuvant treatment modalities influence these recovery patterns. The findings will contribute to a better understanding of post-operative dysphagia in this patient population, informing clinical practice and optimizing patient care.

## Methods

### Study design and participants

A prospective cohort study was conducted at the Department of Otolaryngology – Head and Neck Surgery, University Medical Center (UMC), Ho Chi Minh City, Vietnam, from 6th October 2022–31st March 2025. UMC is a leading public teaching hospital, state-owned and managed, and directly affiliated with the University of Medicine and Pharmacy at Ho Chi Minh City.

The inclusion criteria for the study were as follows: 1) aged 18 years or older; 2) having a previously untreated, pathologically confirmed squamous cell carcinoma of the tongue, floor of the mouth, tonsil, soft palate, or pharyngeal wall; 3) surgery indicated and performed at UMC; 4) alertness and the ability to comprehend auditory instructions; 5) the capability to cooperate throughout the research process; and 6) completion of a full postoperative follow-up schedule for a minimum of 6 months. The exclusion criteria consisted of 1) a previous diagnosis of head and neck cancer at a site different from the surgical area; 2) prior radiation therapy to the head and neck region before surgery; and 3) pre-existing recurrent laryngeal nerve palsy or a medical history linked to diagnosed neuromuscular disorders.

During the study period, all consecutive patients presenting to UMC's Department of Otolaryngology – Head and Neck Surgery, either as new admissions or referrals from other hospitals, with a diagnosis of squamous cell carcinoma of the oral cavity and oropharyngeal region that met all predefined inclusion criteria and for whom surgery was indicated and performed, were considered for enrollment. To optimize sample size and enhance representativeness by including a diverse patient population, we did not conduct a formal sample size calculation; rather, all eligible patients encountered within the study timeframe were enrolled. The study protocol and all procedures were reviewed and approved by the Ethics Committee of the University of Medicine and Pharmacy at Ho Chi Minh City (approval number: 727/HĐĐĐ-ĐHYD). All patients provided informed written consent prior to their enrollment in the study.

### Data collection and pre-operative evaluation

All patients enrolled in the study underwent comprehensive pre-operative evaluations. These assessments included demographic data (age and gender), Body Mass Index (BMI), presence of comorbidities, location of the primary tumor, and clinical TNM staging according to the American Joint Committee on Cancer (AJCC) guidelines. Patients then underwent surgical resection of the tumor, which included neck dissection where indicated, and reconstructive surgery for surgical defects as necessary. Postoperative radiotherapy was indicated for patients with T3-T4 tumors, positive or close surgical margins, perineural tumor spread, multiple positive lymph nodes, or extranodal spread.

### Postoperative swallowing assessment

After surgery, all patients were instructed to begin swallowing rehabilitation therapy as soon as their clinical condition permitted, typically around one week postoperatively. The rehabilitation program included oral cavity care, active movement exercises for the jaw, cheeks, and tongue, as well as therapeutic swallowing postures and maneuvers. Swallowing assessments were performed at 1, 3, and 6 months after surgery. The evaluation focused on several outcomes, including Functional Oral Intake Scale (FOIS) ratings, Swallowing Ability and Safety Scale (SASS), Eating Assessment Tool-10 (EAT-10), Penetration-Aspiration scale (PAS), and the use of compensatory swallowing strategies or postures when swallowing.

- **Functional Oral Intake Scale (FOIS):** This 7-point ordinal scale assesses the level of oral intake, ranging from 1 (nothing by mouth) to 7 (total oral diet with no restrictions). A higher FOIS score (Levels 4–7) indicates improved functional oral intake, less severe dysphagia, and a reduced reliance on alternative feeding methods, correlating with a lower risk of penetration or aspiration. Conversely, lower FOIS scores (Levels 1–3) signify severe dysphagia, often necessitating tube feeding for nutritional support [14,15].

- **Swallowing Ability and Safety Scale (SASS):** This is a clinician-rated tool designed to objectively evaluate swallowing ability, particularly for oral and oropharyngeal cancer patients. SASS utilizes an MTF score, which sums scores (1–5 each) from three subscales: Method of food intake (M), Time for food intake (T), and Group of the food that can be taken (F). An MTF score below 10 (<10) indicates poor swallowing ability, while a score of 10 or more (≥10) signifies gGood ability, aiding in assessing dysphagia severity and treatment effectiveness [10,11].

- **Eating Assessment Tool-10 (EAT-10):** EAT-10 is a self-administered questionnaire where patients rate each of 10 items on a 5-point Likert scale from 0 ("no problem") to 4 ("severe problem"). These questions assess the subjective severity and impact of swallowing difficulties across functional (e.g., weight loss, interference with social eating, effort for liquids/solids/pills), emotional (e.g., pleasure of eating affected, swallowing is stressful), and physical (e.g., pain, food sticking in throat, coughing when eating) aspects of swallowing. The total score is the sum of these 10 items, ranging from 0 to 40, with a higher score indicating more severe dysphagia symptoms. An EAT-10 score of 3 or greater on any single item or a total score of 15 or greater is considered abnormal, suggesting the presence of dysphagia and a need for further evaluation [12,19].

### Fiberoptic endoscopic evaluation of swallowing examination protocol

Penetration and aspiration were assessed using Fiberoptic Endoscopic Evaluation of Swallowing (FEES). During the FEES examination, participants were given three 5-ml bolus trials: moderately thick liquid, extremely thick liquid, and thin liquid, in that order. The moderately thick and extremely thick liquids were prepared by mixing bottled water with Thick-ened Up Clear (Nestlé) powder according to the manufacturer's instructions, and then tinted with blue and green food dyes, respectively [20]. Sterilized fresh milk was used as thin liquid. The tip of a Karl Storz flexible endoscope was positioned just above the epiglottis throughout the evaluation [21,22]. No local anesthetics or vasoconstrictors were administered to the patients' mucosa at any point [18, [23]]. Penetration and aspiration during each trial were assessed using the Penetration-Aspiration Scale (PAS) [24].

### Data analysis

Descriptive statistics were used to summarize patient demographics, tumor characteristics, treatment modalities, and swallowing outcome measures. Continuous variables were presented as means and standard deviations (SD), while categorical variables were reported as frequencies and percentages. To examine changes in swallowing function and oral intake over time, paired comparisons between time points (1 month, 3 months, and 6 months post-surgery) were performed using paired t-tests or McNemar's Chi-squared tests when approriate. To investigate the effect of adjuvant treatment modalities (none, radiotherapy, chemoradiotherapy) on the trajectory of swallowing outcomes while accounting for repeated measurements within individuals over time, Generalized Estimating Equations (GEE) were employed. The model included three main predictors (i.e., time point, adjuvant treatment group, and their interaction term) to examine longitudinal changes in swallowing outcomes and the differential effects across treatment modalities. No additional covariates were entered into the model because the number of outcome events was limited, and the inclusion of further variables could compromise model stability.

Moreover, our primary aim was to characterize recovery trajectories rather than to build a fully adjusted predictive model. The GEE approach, using an exchangeable working correlation structure with robust (sandwich) standard errors, inherently accommodates unbalanced repeated measures and utilizes all available data at each time point. Therefore, missing data were not expected to bias the estimates, and no imputation was performed.From the fitted GEE models, estimated marginal means (for continuous outcomes) and predicted probabilities (for binary outcomes) together with their 95% confidence intervals were computed to quantify effect sizes for each swallowing outcome across treatment groups and time points. Statistical significance was defined at $p < 0.05$, and all analyses were conducted using Stata version 17.

## Results

This study initially enrolled 99 patients with oral cavity/oropharyngeal SCC, with 95 cases proceeding to preoperative evaluation after excluding 4 cases that did not meet eligibility criteria. Following surgery and initial assessments, the cohort for 1-month and 3-month dysphagia assessments remained at 94 cases. Ultimately, after further exclusions, 89 patients completed the 6-month dysphagia assessment and were included in this study (Fig 1). The demographic and clinical characteristics of the study cohort are presented in Table 1.

The study population of 89 patients was predominantly male (80.9%) and under 65 years of age (82.0%), with most patients having a Body Mass Index within the normal or overweight range (89.9%). Tumor characteristics showed T2 stage to be the most common (44.9%), with over half the patients having N0 cervical nodal status (52.8%), and the tongue and floor of mouth as the primary tumor site for the majority (67.4%). Furthermore, the cohort primarily consisted of patients without comorbidities (79.8%) or postoperative oral-pharyngeal infections (86.5%), with about 60% received adjuvant therapy, including radiotherapy or chemoradiotherapy.

### Swallowing and oral intake outcomes over time

Subjective and objective swallowing and oral intake parameters were assessed at 1, 3, and 6 months post-surgery, revealing distinct patterns of change, with are summarized in Table 2. The mean SASS score demonstrated overall improvement from $11.1 \pm 0.4$ at 1 month to $12.0 \pm 0.5$ at 6 months. However, the proportion of patients with poor swallowing ability significantly rose from 12.4% at 1 month to 28.1% at 3 months ($p = 0.002$) before significantly decreasing to 14.6% at 6 months ($p = 0.001$). Similarly, the mean FOIS score significantly improved overall ($p < 0.001$), decreasing slightly from $5.8 \pm 0.1$ at 1 month to $5.3 \pm 0.1$ at 3 months, then markedly increasing to $6.3 \pm 0.1$ at 6 months, indicating substantial dietary recovery. The mean EAT-10 score for perceived dysphagia severity significantly increased from $8.8 \pm 0.8$ at 1 month to $13.5 \pm 0.9$ at 3 months ($p < 0.001$) before significantly decreasing to $8.3 \pm 0.8$ at 6 months ($p < 0.001$), reflecting improved patient perception. This was supported by a reduction in abnormal eating ability assessed by EAT-10 scores from 89.9% at 1 month to 76.4% at 6 months ($p = 0.005$). Concurrently, the rate of penetration/aspiration for thin liquids also significantly increased from 32.6% at 1 month to 44.9% at 3 months ($p = 0.034$), and then significantly decreased to 31.5% at 6 months ($p = 0.014$). For honey-thick liquids, aspiration significantly decreased from 6.7% at 1 month to 0% at 6 months ($p = 0.014$). Correspondingly, the use of compensatory swallowing maneuvers for thin liquids significantly decreased from 14.6% at 1 month to 6.7% at 6 months ($p = 0.020$).

### Estimated changes in swallowing and oral intake scores by adjuvant treatment

Analysis of mean EAT-10 scores (Fig 2A) reveals that while all adjuvant treatment groups experienced a temporary increase in perceived dysphagia severity from 1 month to 3 months, this was followed by a general improvement by 6 months. However, patients who received adjuvant chemoradiotherapy consistently demonstrated higher EAT-10 scores throughout the observed period, suggesting a greater perceived swallowing difficulty compared to patients receiving no adjuvant treatment or adjuvant radiotherapy. Mean SASS scores (Fig 2B) generally increased from 1 month to 6 months across all adjuvant treatment groups, indicating an overall improvement in objective swallowing ability. Similarly, mean FOIS scores (Fig 2C) consistently demonstrated an upward trajectory from 1 month to 6 months across all groups, reflecting a progressive improvement in functional oral intake. Patients receiving adjuvant chemoradiotherapy appear to have consistently lower FOIS scores across the time points, suggesting a greater impact on functional oral intake.

### Estimated changes in penetration/Aspiration and use of compensatory maneuvers

Fig 3 illustrates changes in penetration/aspiration rates and the use of compensatory swallowing maneuvers, categorized by liquid consistency and adjuvant treatment group.

 

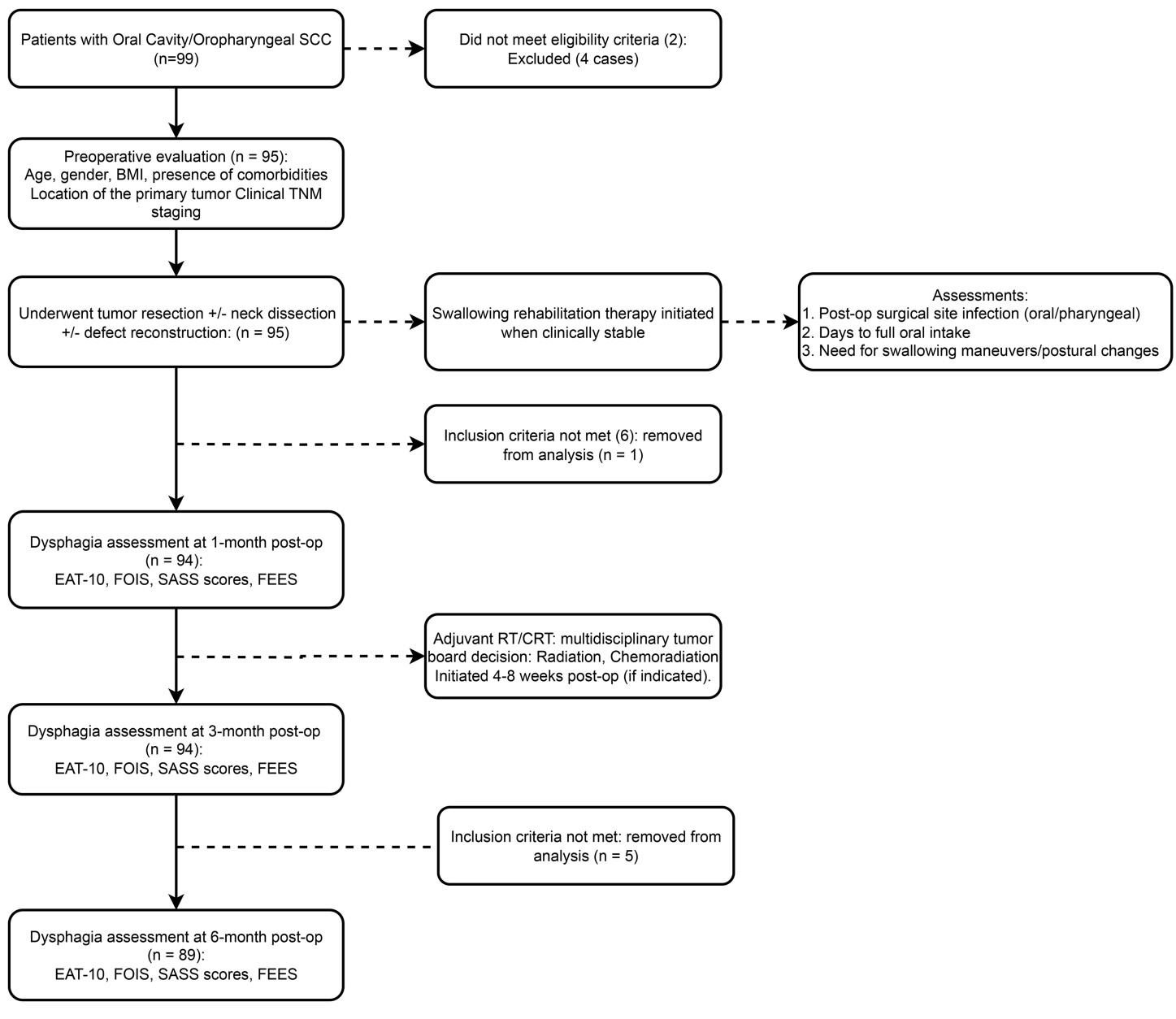

**Fig 1. Study flowchart.**

For moderately thick liquids (Fig 3A) and extremely thick liquids (Fig 3C), the rates of penetration/aspiration showed some variation across different time points and adjuvant treatment groups, without a clear, consistent pattern of significant increase or decrease observed. In contrast, for thin liquids (Fig 3E), penetration/aspiration rates generally reached their highest point around 3 months post-surgery. This was most noticeable in groups that received adjuvant therapy. After this peak, the rates tended to decline by 6 months. Patients who underwent adjuvant chemoradiotherapy consistently demonstrated higher rates of penetration/aspiration for thin liquids (Fig 3E).

The patterns for the use of compensatory maneuvers generally mirrored the trends observed for penetration/aspiration across all liquid consistencies. Reliance on these maneuvers tended to be higher at 3 months post-surgery, especially for

**Table 1. General characteristics of the study population and common factors affecting outcome (n = 89).**

| Characteristics | Frequency | Percentage |
| --- | --- | --- |
| **Age** *(year)* | | |
| <65 | 73 | 82.0 |
| ≥65 | 16 | 18.0 |
| **Sex** | | |
| Male | 72 | 80.9 |
| Female | 17 | 19.1 |
| **Body Mass Index** *(kg/m²)* | | |
| <18.5 | 9 | 10.1 |
| 18.5-<23 | 42 | 47.2 |
| ≥23 | 38 | 42.7 |
| **Tumor T stage** | | |
| T1 | 13 | 14.6 |
| T2 | 40 | 44.9 |
| T3 | 24 | 27.0 |
| T4 | 12 | 13.5 |
| **Cervical Nodal Stage (N)** | | |
| N0 | 47 | 52.8 |
| N1 | 17 | 19.1 |
| N2 | 22 | 24.7 |
| N3 | 3 | 3.4 |
| **Comorbidity** | | |
| Yes | 18 | 20.2 |
| No | 71 | 79.8 |
| **Postoperative Oral-Pharyngeal Infection** | | |
| Yes | 12 | 13.5 |
| No | 77 | 86.5 |
| **Adjuvant Therapy** | | |
| No | 36 | 40.4 |
| Adjuvant radiotherapy | 27 | 30.3 |
| Adjuvant chemoradiotherapy | 26 | 29.2 |
| **Primary tumor site** | | |
| Tongue and floor of mouth | 60 | 67.4 |
| Tonsil | 21 | 23.6 |
| Posterior pharyngeal wall | 5 | 5.6 |
| Soft palate | 2 | 2.2 |
| Base of tongue | 1 | 1.1 |

thin liquids (Fig 3F). Patients in the adjuvant chemoradiotherapy group consistently reported a higher and more sustained need for compensatory maneuvers across all liquid consistencies (Fig 3B, 3D, and 3F). This suggests that these patients experienced greater underlying functional impairment, thereby requiring them to use these adaptive strategies more frequently to swallow safely.

## Discussion

This prospective cohort study provides valuable insights into the longitudinal changes in swallowing and oral intake outcomes during the first 6 months following surgical treatment for oral cavity and oropharyngeal cancers, particularly

**Table 2. Swallowing and oral intake outcomes over time.**

| Characteristics | Follow-up after the surgery | | | p 3 months vs 1 month | p 6 months vs 1 month | p 6 months vs 3 months |
|---|---|---|---|---|---|---|
| | 1 month | 3 months | 6 months | | | |
| **SASS score** M (SD) | 11.1 (1.8) | 11.2 (2.3) | 12.0 (2.1) | 0.736 | **<0.001** | **<0.001** |
| **SASS score** | | | | | | |
| Poor (<10) | 11 (12.4) | 25 (28.1) | 13 (14.6) | **0.002** | 0.637 | **0.001** |
| Good (≥ 10) | 78 (87.6) | 64 (71.9) | 76 (85.4) | | | |
| **FOIS score** M (SD) | 5.8 (1.1) | 5.3 (1.5) | 6.3 (0.7) | **<0.001** | **<0.001** | **<0.001** |
| **EAT-10 score** M (SD) | 8.8 (5.6) | 13.5 (9.7) | 8.3 (6.7) | **<0.001** | 0.552 | **<0.001** |
| **EAT-10 score** | | | | | | |
| Normal (<3) | 9 (10.1) | 17 (19.1) | 21 (23.6) | **0.005** | **<0.001** | **0.046** |
| Abnormal (≥3) | 80 (89.9) | 72 (80.9) | 68 (76.4) | | | |
| **Penetration or Aspiration** | | | | | | |
| Nectar-thick liquid | 13 (14.6) | 14 (15.7) | 8 (9.0) | 0.782 | 0.132 | 0.133 |
| Honey-thick liquid | 6 (6.7) | 7 (7.9) | 0 (0) | 0.705 | **0.014** | **0.008** |
| Thin liquid | 29 (32.6) | 40 (44.9) | 28 (31.5) | **0.034** | 0.857 | **0.014** |
| **Use of Swallowing Maneuver** | | | | | | |
| Nectar-thick liquid | 9 (10.2) | 14 (15.7) | 7 (7.9) | 0.157 | 0.083 | **0.008** |
| Honey-thick liquid | 10 (11.2) | 13 (14.6) | 7 (7.9) | 0.257 | 0.083 | **0.034** |
| Thin liquid | 13 (14.6) | 16 (18.0) | 6 (6.7) | 0.366 | **0.020** | **0.002** |

highlighting the differential impact of adjuvant therapies. Our findings demonstrate a complex recovery trajectory, characterized by an initial post-operative decline in perceived and objective swallowing function at 3 months, followed by a general improvement by 6 months. Crucially, patients receiving adjuvant chemoradiotherapy consistently experienced more severe dysphagia across various outcome measures.

The observed pattern of an initial worsening of swallowing function around 3 months, followed by recovery by 6 months, is a critical finding. The mean EAT-10 and FOIS scores, as well as the rates of thin liquid penetration/aspiration and compensatory maneuver use, all exhibited a temporary deterioration at 3 months post-surgery before showing signs of improvement. This acute phase of increased dysphagia severity often correlates with the peak of treatment-related toxicities, such as mucositis, pain, and fatigue, which are common after surgical recovery and the onset of adjuvant therapy [3,25]. Our finding of a dynamic swallowing trajectory, marked by a transient dip around the 3-month mark corresponding to the peak of treatment toxicities, aligns with observations from other prospective studies on head and neck cancer. A five-year prospective study specifically on oral cancer treatment highlighted that swallowing duration and frequency demonstrate longitudinal changes that often do not return to pre-treatment levels, emphasizing the long-term sequelae. The addition of adjuvant radiotherapy or chemoradiotherapy significantly increases the risk of impaired swallowing function, underscoring the necessity of closely tracking outcomes in these subgroups [18]. While our study assesses outcomes from 1 month post-surgery, this 3-month function aligns with findings from other studies that report significant declines in swallowing during or shortly after chemoradiotherapy, with subsequent partial recovery [3,25–27]. For instance, Patterson et al. (2013) noted a marked deterioration in MDADI scores (similar to EAT-10) from pre-treatment to 3 months post-chemoradiotherapy, though their study suggested limited improvement thereafter, which contrasts with the more robust recovery by 6 months observed in our EAT-10 and FOIS results [27]. This divergence may reflect differences in patient cohorts, specific treatment protocols, or rehabilitation adherence, emphasizing the dynamic nature of recovery. Chiu et al. (2022) also reported that swallowing dysfunction changes over time, with the worst scores often seen around 3 months post-treatment, consistent with our observations [8].

  

**A) EAT score**

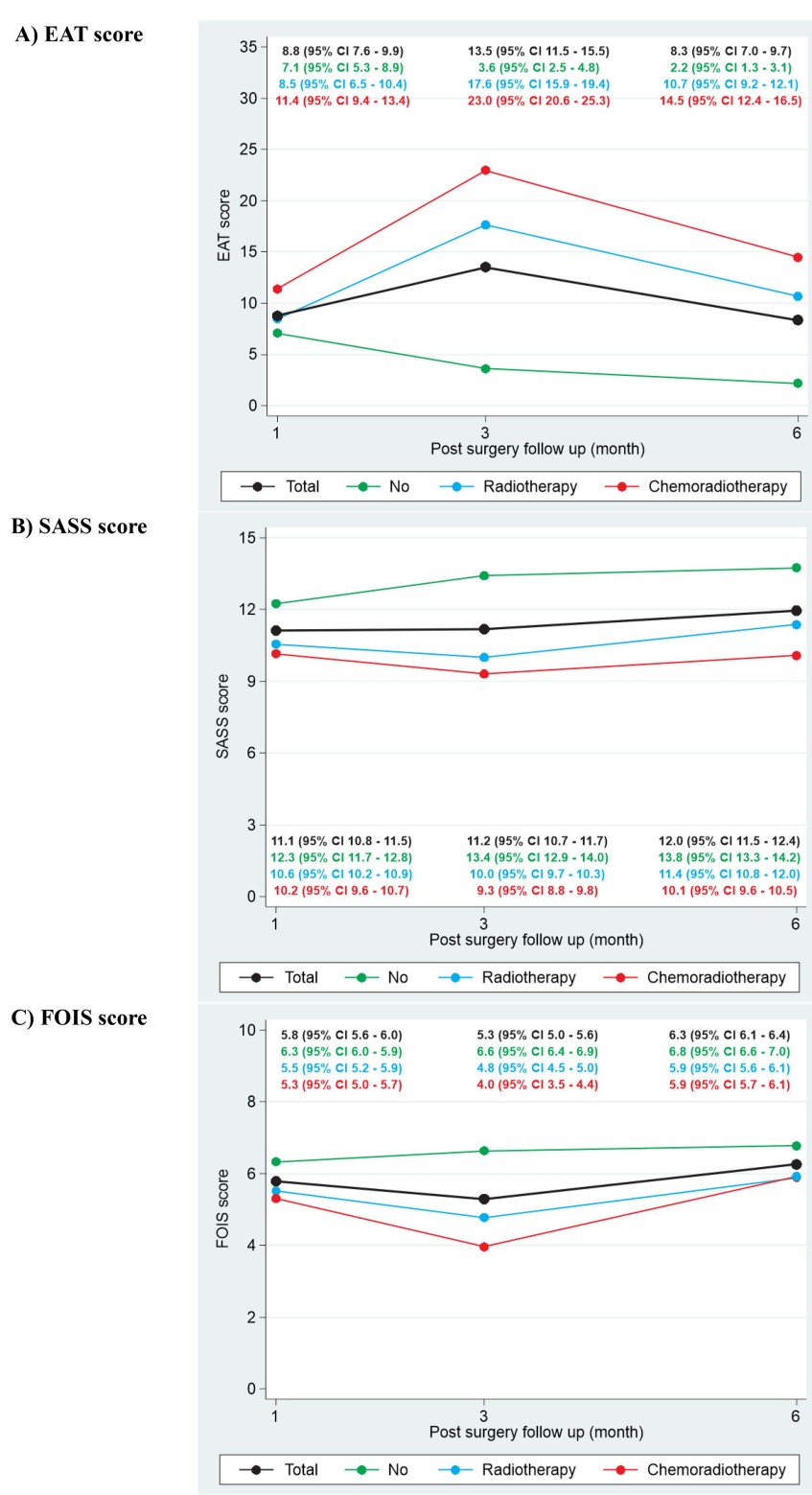

**B) SASS score**

**C) FOIS score**

**Fig 2. Estimated changes in swallowing and oral intake scores during follow-up, stratified by adjuvant treatment.**

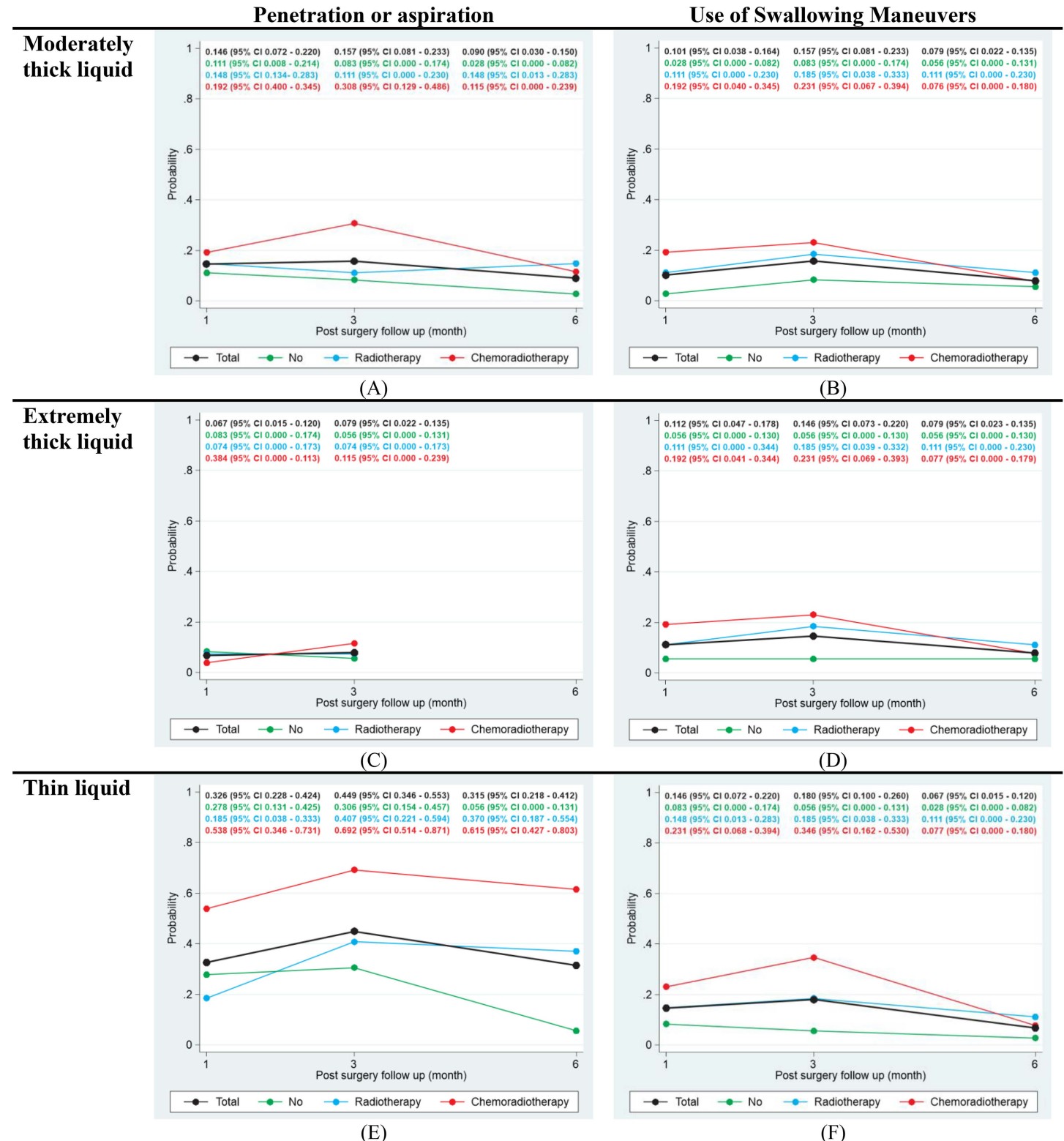

**Fig 3. Estimated changes in penetration/Aspiration and use of compensatory maneuvers during follow-up, stratified by adjuvant treatment.**

Our objective measures (SASS, FOIS, penetration/aspiration) and patient-reported outcomes (EAT-10) generally followed a similar trend of initial worsening followed by recovery. We noted a transient and seemingly paradoxical increase in the proportion of patients with 'poor SASS' scores (<10) at the 3-month time point, despite an overall improvement in the mean SASS score. This may be attributed to the heterogeneity of the cohort and the acute effects of adjuvant (chemo) radiotherapy that typically commence or peak around this period, causing temporary, severe impairment in a subset of patients (e.g., due to mucositis or pain), thus disproportionately shifting the distribution towards greater severity. This highlights the importance of using multiple assessment paradigms, as patient perception (EAT-10) and objective measures (SASS, FOIS) may not always perfectly correlate [3,13], though they often show significant relationships [28,29]. The observed magnitude of change in EAT-10 scores in the overall cohort (e.g., a decrease from 13.5 at 3 months to 8.3 at 6 months) surpasses the generally accepted Minimal Clinically Important Difference (MCID) for EAT-10, which is often cited as a change of 2-3 points [12,19]. Similarly, the changes observed in FOIS scores, particularly the transition from level 5 (total oral diet with multiple consistencies) at 3 months to level 6 (total oral diet with single restriction) at 6 months, represent a clinically significant step toward less restriction. This interpretation is supported by studies demonstrating that the necessary EAT-10 cut-off points for identifying unsafe swallowing vary dramatically depending on the treatment phase (e.g., from 3 at baseline to 15 post-CRT) [30], and that the EAT-10 has a strong correlation with objective functional intake measured by the FOIS [31]. Our higher penetration/aspiration rates for thin liquids (peaking at 44.9% at 3 months) compared to some literature (e.g., around 28% reported by Patterson et al. at 3 months post-chemoradiation therapy) may reflect the specific patient population, surgical extent, or assessment protocols [27]. The increased use of compensatory maneuvers coinciding with peak penetration/aspiration rates further underscores patients' adaptive strategies to maintain oral intake when faced with functional deficits [32]. Studies have shown that early detection of swallowing problems and timely speech and language therapy interventions can aid adaptation and promote better outcomes, emphasizing the role of rehabilitation in mitigating these deficits [25,33].

The pronounced impact of chemoradiotherapy on long-term functional outcomes is a consistent finding in the literature, reinforcing our results that this group experiences more severe and prolonged dysphagia. A systematic review on tube feeding in HNC patients undergoing CRT, for instance, showed that while Percutaneous Endoscopic Gastrostomy (PEG) and Nasogastric Tube (NGT) tubes are comparable for nutrition outcomes and survival, there is increasing evidence suggesting higher risk of long-term swallow dysfunction associated with PEG use [34]. Furthermore, the need for Gastrostomy tube placement itself is associated with pretreatment risk factors unique to RT/CRT cohorts, such as pretreatment dysphagia, low BMI, and advanced tumor stage [35]. This highlights the critical need for early, risk-stratified intervention and counseling in patients receiving combined therapy.

A significant finding of our study is the consistent and more pronounced impact of adjuvant chemoradiotherapy on swallowing outcomes. Patients receiving adjuvant chemoradiotherapy consistently exhibited higher EAT-10 scores (greater perceived difficulty), lower FOIS scores (more impacted functional intake), and persistently higher rates of thin liquid penetration/aspiration, coupled with a more sustained need for compensatory maneuvers. This aligns strongly with existing literature demonstrating that the addition of chemotherapy to radiotherapy significantly exacerbates dysphagia severity and long-term swallowing morbidity due to increased acute and late toxicities like mucositis and fibrosis [3,7,8,25,26,36,37]. The combined insult of surgery and chemoradiotherapy represents a substantial functional challenge, often leading to more complex and protracted rehabilitation needs and potentially higher rates of feeding tube dependence [38].

While our study provides valuable longitudinal data, it has several limitations. First, as a single-center study, the generalizability of our findings may be limited, and variations in surgical techniques, reconstruction approaches, and adjuvant protocols across institutions could influence outcomes.

Second, the study employed a pragmatic consecutive sampling approach rather than a formal sample size calculation, which limits the statistical power for more granular subgroup analyses and may affect the generalizability of the findings.

. Future studies should aim for larger, multi-center cohorts to explore these nuances. Third, in this study, the follow-up period was restricted to 6 months. While this duration captures the critical phase of acute recovery and the effects of adjuvant therapy, it is important to acknowledge that clinically significant late dysphagia often manifests or progresses beyond 12 months post-treatment. Therefore, additional studies with extended follow-up are warranted to better evaluate long-term functional changes.[16,25,27]. Fourth, a key limitation is the heterogeneous nature of the cohort, combining both oral cavity and oropharyngeal cancers. These subsites have distinct surgical approaches and functional outcomes (e.g., free flap reconstruction vs. primary closure), which are known to significantly influence swallowing recovery. Our current analysis does not provide the statistical power to separately model the longitudinal effects for each specific tumor subsite or reconstruction type, representing a potential confounding factor in the observed trajectory results. Lastly, although we utilized validated objective and subjective swallowing assessments, a direct comparison of the individual effects of specific surgical defect characteristics on the longitudinal trajectories of swallowing outcomes by adjuvant treatment could not be performed in this study. Future research could also explore the impact of specific tumor characteristics (e.g., HPV status for oropharyngeal cancer), and surgical defect extension on swallowing outcomes, as this is an evolving area of interest [39].

## Conclusion

Our study underscores the dynamic nature of swallowing recovery after oral cavity and oropharyngeal cancer surgery, highlighting a temporary functional decrease at 3 months followed by general improvement by 6 months. Moreover, adjuvant chemoradiotherapy is associated with a more significant and sustained burden of dysphagia. These findings emphasize the critical need for comprehensive, multidisciplinary care, including proactive swallowing rehabilitation and tailored patient counseling, especially for those undergoing adjuvant chemoradiotherapy, to optimize functional outcomes and improve quality of life.

## Supporting information

**S1 File. Data and dictionary.**
(ZIP)

## Acknowledgments

We thank all the data collectors for their valuable contributions to this study. The authors are also grateful for the support from Department of Otolaryngology – Head and Neck Surgery, University Medical Center (UMC) Ho Chi Minh City and to all the participants who contributed to it.

## Author contributions

**Conceptualization:** Loan Thi Hong Nguyen, Xuan Quang Ly.

**Data curation:** Loan Thi Hong Nguyen, Duc Tan Vo, Xuan Quang Ly.

**Formal analysis:** Truc Thanh Thai.

**Investigation:** Loan Thi Hong Nguyen, Duc Tan Vo.

**Methodology:** Truc Thanh Thai, Xuan Quang Ly.

**Software:** Truc Thanh Thai.

**Supervision:** Xuan Quang Ly.

**Validation:** Truc Thanh Thai.

**Visualization:** Truc Thanh Thai.

**Writing – original draft:** Loan Thi Hong Nguyen, Duc Tan Vo, Truc Thanh Thai, Xuan Quang Ly.

**Writing – review & editing:** Loan Thi Hong Nguyen, Duc Tan Vo, Truc Thanh Thai, Xuan Quang Ly.

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
