## [Decision Letter · Decision Letter 0]

13 Oct 2025

Dear Dr. Ly,

Thank you for submitting your manuscript to PLOS ONE. After careful consideration, we feel that it has merit but does not fully meet PLOS ONE’s publication criteria as it currently stands. Therefore, we invite you to submit a revised version of the manuscript that addresses the points raised during the review process.

**ACADEMIC EDITOR: The article needs a major overhaul.**

We look forward to receiving your revised manuscript.

Kind regards,

Satyajeet Rath

Academic Editor

PLOS ONE

Journal Requirements:

2. Please remove your figures from within your manuscript file, leaving only the individual TIFF/EPS image files, uploaded separately. These will be automatically included in the reviewers’ PDF.

3. Please include captions for your Supporting Information files at the end of your manuscript, and update any in-text citations to match accordingly. Please see our Supporting Information guidelines for more information: http://journals.plos.org/plosone/s/supporting-information .

Additional Editor Comments:

There are some major issues in the article which the authors need to address

Reviewers' comments:

Reviewer's Responses to Questions

**Comments to the Author**

1. Is the manuscript technically sound, and do the data support the conclusions?

Reviewer #1: Yes

Reviewer #2: Yes

2. Has the statistical analysis been performed appropriately and rigorously?

Reviewer #1: Yes

Reviewer #2: Yes

3. Have the authors made all data underlying the findings in their manuscript fully available?

Reviewer #1: Yes

Reviewer #2: Yes

4. Is the manuscript presented in an intelligible fashion and written in standard English?

Reviewer #1: Yes

Reviewer #2: Yes

Reviewer #1: This is a well-designed prospective cohort study assessing longitudinal swallowing outcomes in patients with oral cavity and oropharyngeal squamous cell carcinoma following surgery, with stratification by adjuvant therapy (none, radiotherapy, chemoradiotherapy). The study is clinically relevant, addresses a gap in the literature, and is presented in a clear format with well-structured sections. The inclusion of both patient-reported outcomes (EAT-10) and objective assessments (SASS, FOIS, FEES) is a major strength, offering a comprehensive evaluation of swallowing recovery. However, while the study is promising, several aspects require clarification, refinement, and additional discussion before the manuscript can be considered for publication.

Major Issues

• No sample size calculation was performed; the rationale for including “all consecutive eligible patients” is acceptable but should be explicitly discussed as a limitation regarding statistical power and generalizability.

• The follow-up is restricted to 6 months; late dysphagia beyond this time is clinically important. Authors should acknowledge that long-term swallowing impairment (often >12 months) may differ significantly.

• The use of GEE models is appropriate, but more details are needed. Were covariates included in the model? How was missing data (if any) handled?

• Results are presented mostly descriptively; effect sizes (e.g., mean differences with 95% CI) would strengthen the interpretation.

• The cohort includes a mix of oral cavity and oropharyngeal cancers with different prognostic and functional implications. The manuscript would benefit from subgroup reporting or, at minimum, a discussion of how tumor subsite and reconstructive procedures might influence swallowing recovery.

• The paradoxical finding that SASS mean scores improved while the proportion of “poor SASS” patients increased at 3 months should be better explained, possibly by subgroup variability or skewed distribution.

• The clinical relevance of changes in EAT-10 and FOIS should be discussed.

Minor Issues

• Some phrasing could be improved for fluency. Minor grammatical errors and typographical inconsistencies should be corrected.

• Literature review is adequate but could be updated with recent systematic reviews and large cohort studies (2021–2024) focusing on swallowing outcomes post-surgery and CRT. Add references on clinically meaningful thresholds for EAT-10 and FOIS.The abstract is informative but somewhat lengthy. It could be more concise, emphasizing key results and clinical implications.

Reviewer #2: Minor suggestion to change "due to failed eligibility" to "did not meet eligibility criteria" . Similarly suggest changing "disqualified" to removed from analysis as did not meet eligibility criteria.

**Do you want your identity to be public for this peer review?** For information about this choice, including consent withdrawal, please see our Privacy Policy

Reviewer #1: No

Reviewer #2: No

---

## [Author Response · Author response to Decision Letter 1]

24 Nov 2025

Please see our responses in the file attached. Thank you very much.

---

## [Decision Letter · Decision Letter 1]

4 Jan 2026

Postoperative swallowing recovery in oral and oropharyngeal cancer: A prospective analysis of functional changes and adjuvant therapy effects

PONE-D-25-43584R1

Dear Dr. Ly,

We’re pleased to inform you that your manuscript has been judged scientifically suitable for publication and will be formally accepted for publication once it meets all outstanding technical requirements.

Kind regards,

Satyajeet Rath

Academic Editor

PLOS One

Additional Editor Comments (optional):

All the suggested changes and desired comments have been positively addressed by the authors.

Reviewers' comments:

Reviewer's Responses to Questions

**Comments to the Author**

Reviewer #1: All comments have been addressed

2. Is the manuscript technically sound, and do the data support the conclusions?

Reviewer #1: Yes

3. Has the statistical analysis been performed appropriately and rigorously?

Reviewer #1: Yes

4. Have the authors made all data underlying the findings in their manuscript fully available?

Reviewer #1: Yes

5. Is the manuscript presented in an intelligible fashion and written in standard English?

Reviewer #1: Yes

Reviewer #1: The authors have appropriately addressed the reviewers’ concerns. Remaining issues are minor and editorial, rather than conceptual or methodological.

**Do you want your identity to be public for this peer review?** For information about this choice, including consent withdrawal, please see our Privacy Policy

Reviewer #1: No

---

## [Editor Report · Acceptance letter]

PONE-D-25-43584R1

PLOS One

Dear Dr. Ly,

I'm pleased to inform you that your manuscript has been deemed suitable for publication in PLOS One. Congratulations! Your manuscript is now being handed over to our production team.

Kind regards,

on behalf of

Dr. Satyajeet Rath

Academic Editor

PLOS One